# Reusability Report: Evaluating the performance of a meta-learning foundation model on predicting the antibacterial activity of natural products

Caitlin M. Butt [1] & Allison S. Walker [1,2,3] ✉

Deep learning foundation models are becoming increasingly popular for use in bioactivity prediction. Recently, Feng et al. developed ActFound, a bioactive foundation model that jointly uses pairwise learning and meta-learning. By utilizing these techniques, the model is capable of being fine-tuned to a more specific bioactivity task with only a small amount of new data. Here, to investigate the generalizability of the model, we looked to fine-tune the foundation model on an antibacterial natural products (NPs) dataset. Large, labelled NPs datasets, which are needed to train traditional deep learning methods, are scarce. Therefore, the bioactivity prediction of NPs is an ideal task for foundation models. We studied the performance of ActFound on the NPs dataset using a range of few-shot settings. Additionally, we compared ActFound's performance with those of other state-of-the-art models in the field. We found ActFound was unable to reach the same level of accuracy on the antibacterial NPs dataset as it did on other cross-domain tasks reported in the original publication. However, ActFound displayed comparable or better performance compared to the other models studied, especially at the low-shot settings. Our results establish ActFound as a useful foundation model for the bioactivity prediction of tasks with limited data, particularly for datasets that contain the bioactivities of similar compounds.

The bioactivity of compounds plays a key role in drug discovery. Bioactivity refers to the effect, either beneficial or adverse, a compound has on a biological process[1]. It encompasses the efficacy, potency and selectivity of a compound and is important in the identification of hits in a drug campaign and subsequent lead optimization[2]. Deep learning (DL) approaches have shown promise in their ability to predict the bioactivity of compounds[3–7]. However, DL models require large, high-quality datasets to accurately identify patterns within the data[8]. Many bioactivity tasks do not have an adequate number of labelled data sufficient for training. One solution is to use foundation models,

which are pretrained on large, general datasets. These models serve as a 'foundation' that can be fine-tuned for more specific tasks[9].

Recently, Feng et al. introduced ActFound[10], a meta-learning foundation model trained to predict the bioactivity of compounds. To accomplish this, ActFound jointly utilizes meta-learning and pairwise learning. Meta-learning, or 'learning to learn', is a commonly used algorithm to develop foundation models[11,12]. Meta-learning models are trained on a variety of tasks with the intention of creating a model that can quickly adapt to new tasks from only a small amount of new data[13]. By using meta-learning, Feng et al.[10] were able to pretrain their model

[1]Department of Chemistry, Vanderbilt University, Nashville, TN, USA. [2]Department of Biological Sciences, Vanderbilt University, Nashville, TN, USA. [3]Department of Pathology, Microbiology, and Immunology, Vanderbilt University Medical Center, Nashville, TN, USA. ✉e-mail: allison.s.walker@vanderbilt.edu

**Fig. 1 | Overview of the fine-tuning procedure.** Growth-inhibitory assays were used to fine-tune ActFound and create bacteria-specific models.

A *t*-distributed stochastic neighbour embedding (*t*-SNE) comparing the compounds within the NPs dataset and the ChEMBL training data shows overlap, indicating the two datasets likely contain similar compounds (Fig. 2e). To determine the extent of the overlap, we computed the Tanimoto similarities between compounds in the two datasets and found 248 identical compounds (Supplementary Fig. 1). We also analysed how molecular properties, specifically molecular weight, calculated LogP, total polar surface area, hydrogen bond donors, hydrogen bond acceptors and negative log of bioactivity, compare between NPs and the ChEMBL and BindingDB datasets and found that although the distribution differs for some properties, the NP distribution always overlaps the distributions for the other datasets (Supplementary Table 1 and Supplementary Fig. 2). We acknowledge that overlapping compounds between the fine-tuning and training datasets could have caused data leakage that inflated the performance of the model pretrained on ChEMBL assays. Given that this dataset was the result of manual literature curation, it is unlikely that the exact assays in this dataset were deposited into ChEMBL. We investigated the overlap between the bioactivities in the NPs dataset and the ChEMBL database and identified 324 instances where identical molecules were tested in similar assays across the two datasets. When fine-tuning, we found that removing these bioactivities from the dataset had no significant impact on the performance of ActFound (Supplementary Fig. 3). Due to this, we chose to include the overlapping bioactivities in the fine-tuning dataset. In addition, considering that the NPs dataset contained only growth-inhibitory assays, there should be no identical assays in the BindingDB training dataset.

When fine-tuning, we considered each bacterial strain to be its own assay and assessed the performance of ActFound across a range of shot settings. This included using 8–128 fine-tuning compounds as well as using 20–80% of the compounds within each assay for fine-tuning. Averaging the $r^2$ value across all shot settings for ActFound and Act-Found Transfer, a transfer learning variant of ActFound, showed that overall ActFound Transfer had a higher $r^2$ value than ActFound on the NPs dataset (Fig. 2a). However, ActFound was found to have the lowest root mean square error (RMSE) value (Fig. 2b). When looking at the performance for each shot setting, ActFound and ActFound Transfer performed the best in the 16-shot setting, with performance dropping as the shot setting increased (Fig. 2c and Supplementary Fig. 4). This is in contrast with the original publication, where the performance of ActFound increased with the number of compounds used for fine-tuning. Because providing more compounds for fine-tuning should inherently improve performance, we investigated using a percentage of the assays for fine-tuning instead of supplying a specific shot setting. In this case, ActFound performed as expected, with the $r^2$ values increasing as the percentage of compounds used for fine-tuning increased (Fig. 2d). We attribute this behaviour to the fact that only four assays had enough compounds to be used for fine-tuning in the 64- and 128-shot settings. When looking at the performance on each assay, the four largest assays (*Bacillus subtilis*, *Escherichia coli*, *Staphylococcus aureus* and *S. aureus* (MRSA)) yielded the worst model performance (Fig. 3a). This caused the average performance of ActFound across the assays to decrease as the shot setting increased. The performance of ActFound on the four largest assays across the 8- to 128-shot settings showed that the $r^2$ value did increase as the shot setting increased, following the expected trend (Supplementary Fig. 5).

ActFound was found to have varying degrees of performance across the 14 assays, with average $r^2$ values ranging from 0.01 to 0.13. For reference, in the cross-domain setting, the original publication found two kinase inhibitor datasets to have average $r^2$ values between 0.15 and 0.25, so the model performs only slightly worse on some of the NP antibacterial datasets and much worse on others. We also examined fine-tuning the model on a scaffold split, which was defined as a 'realistic split' by Feng et al.[10]. In a scaffold split, the molecules within each assay were split so that the compounds within the fine-tuning

on a wide range of diverse assays, leveraging the information to develop a general foundation model capable of few-shot learning. Pairwise learning was used to address the incompatibility of information within training assays differing in metrics, units and value ranges. Instead of directly predicting the bioactivity of compounds, pairwise learning allows ActFound to predict the difference in bioactivity between two compounds within the same assay[14]. During the fine-tuning stage, ActFound utilized the algorithm *k*-nearest neighbours model-agnostic meta-learning (kNN-MAML), which identifies assays within the training set that are similar to the fine-tuning assay. Leveraging information from similar assays allows for rapid fine-tuning to a new, unseen assay.

In this Reusability Report, we looked to study ActFound's performance on a natural products (NPs) dataset that contains plant-derived compounds with antibacterial activity[15]. NPs are an abundant source of antibiotics, with many approved antibiotics being NPs or NP derivatives[16]. However, antibacterial NPs have historically been plagued by the need for a dereplication process, to avoid frequent rediscovery of known NPs[17]. The lack of new antibiotics being identified has led to a rising interest in DL models for antibacterial activity prediction[18–21]. However, there is a lack of large, labelled bioactivity datasets in the NPs field[22]. This makes it an ideal task for a pairwise meta-learning model like ActFound. In this study, we fine-tuned both the ActFound model pretrained on assays from ChEMBL[23] as well as the model pretrained on assays from BindingDB[24]. We investigated the use of the few-shot setting, fine-tuning the models on differing numbers of NPs within the dataset. The shot settings ranged between 8 and 128 fine-tuning compounds (Fig. 1). We then compared the performance of the ActFound fine-tuned models with other conventional meta-learning models: MAML and ProtoNet as well as transfer learning variants of ActFound and MAML[13,25].

## Fine-tuning ActFound on an NPs dataset

To investigate ActFound's ability to generalize to new domains not explored in the original publication, we fine-tuned the model on an antibacterial NPs dataset. This dataset was curated by Porras et al. in an extensive literature review spanning from 2012 to 2019 and contains the growth-inhibitory activity of NPs against a range of bacteria.

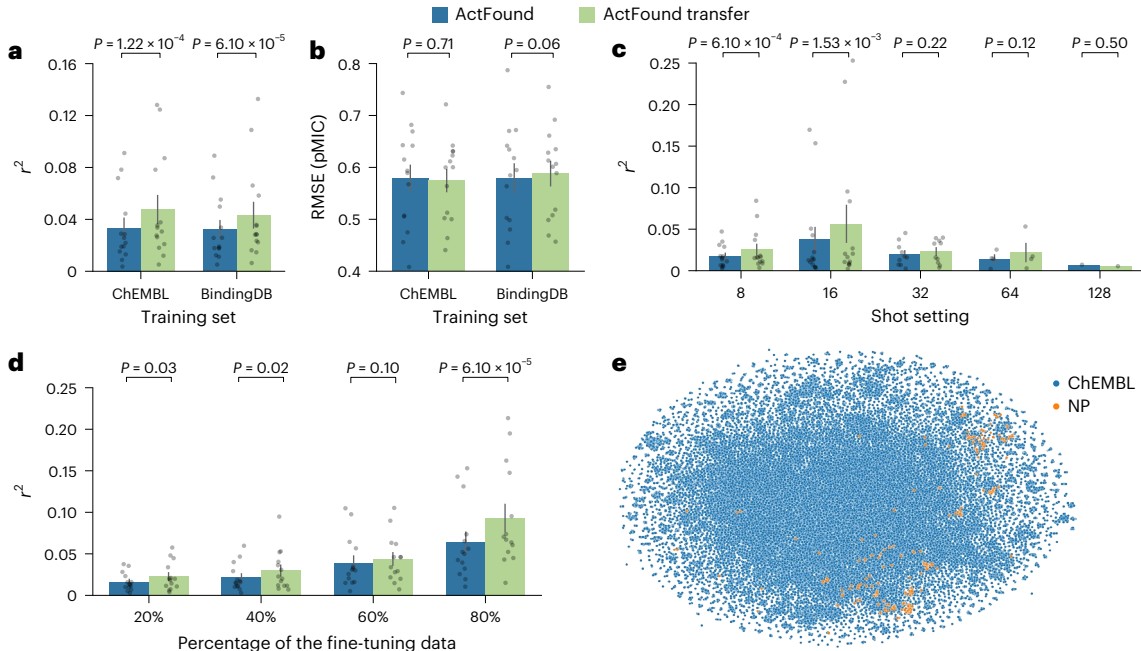

**Fig. 2 | Performance of ActFound on the NPs dataset. a,b**, Bar plots comparing ActFound and ActFound Transfer's performance on the NPs dataset in terms of (**a**) $r^2$ and (**b**) RMSE. Plots show the mean ± standard error of the mean (s.e.m.) performance values across the $n = 14$ assays used to fine-tune the ChEMBL pretrained models and the BindingDB pretrained models. Before plotting, the performance values for the assays were averaged across each shot setting used to fine-tune the models. **c,d**, Bar plots comparing the models' performances in terms of $r^2$ at the (**c**) 8- ($n = 14$), 16- ($n = 14$), 32- ($n = 10$), 64- ($n = 4$) and 128- ($n = 1$)

shot settings and when (**d**) 20% ($n = 14$), 40% ($n = 14$), 60% ($n = 14$) and 80% ($n = 14$) of the assays were used for fine-tuning. Plots show the mean ± s.e.m. performance values across the $n$ number of assays used to fine-tune the pretrained models. Statistical significance between the performance of ActFound and ActFound Transfer was assessed using a one-sided Wilcoxon test. **e**, $t$-SNE plot comparing the molecules in the NPs dataset and a random selection of 50% of the ChEMBL training dataset.

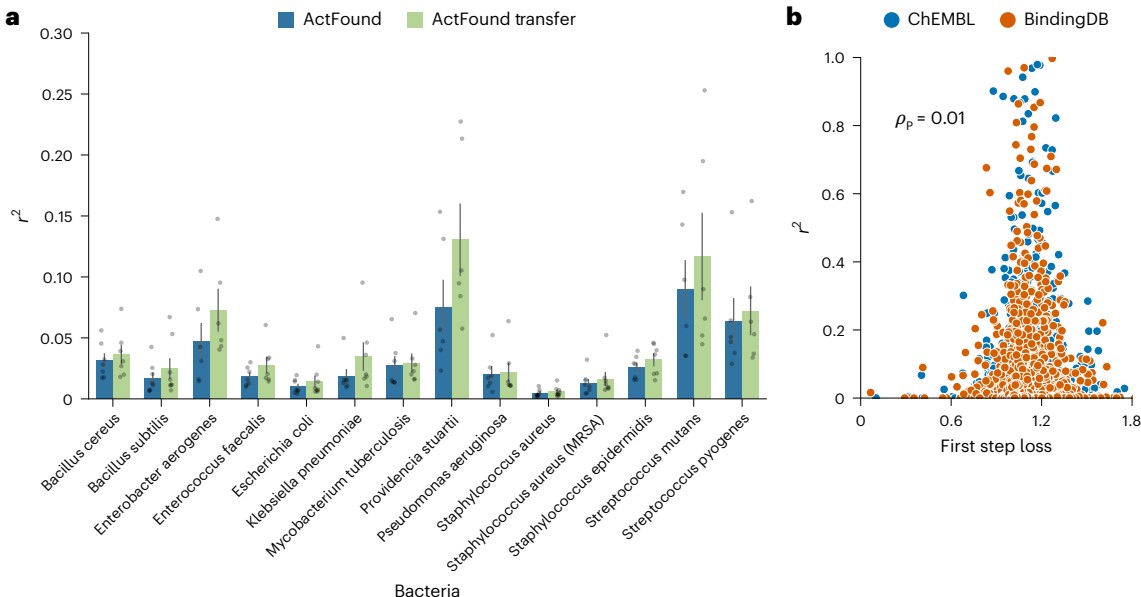

**Fig. 3 | Performance of ActFound on each assay. a**, Bar plot comparing ActFound and ActFound Transfer's performance on each assay in the NPs dataset in terms of $r^2$. Plot shows the mean ± s.e.m. performance value across the $n$ number of shot settings used to fine-tune the ChEMBL pretrained models and the BindingDB pretrained models. The $n$ value for each assay is as follows: *E. aerogenes, P. stuartii, S. mutans, S. pyogenes, $n = 6$; B. cereus, E. faecalis, K. pneumoniae,*

*M. tuberculosis, P. aeruginosa, S. epidermidis, $n = 7$; B. subtilis, E. coli, S. aureus* (MRSA), $n = 8$; *S. aureus, $n = 9$.* **b**, Scatter plot comparing the first step loss and $r^2$ value for every fine-tuning iteration for each assay for the ChEMBL pretrained ActFound and the BindingDB pretrained ActFound. $\rho_P$ is the Pearson correlation between $r^2$ and the first step loss.

set were dissimilar to those in the testing set. We found there to be no significant difference in $r^2$ values for ActFound or ActFound Transfer when a scaffold split was performed instead of a random split (Supplementary Fig. 6).

ActFound utilizes pairwise learning and works under the assumption that similar compounds will have similar bioactivities. This is typically advantageous, as many assay datasets are assembled to investigate structure–activity relationships (SARs), and thus the

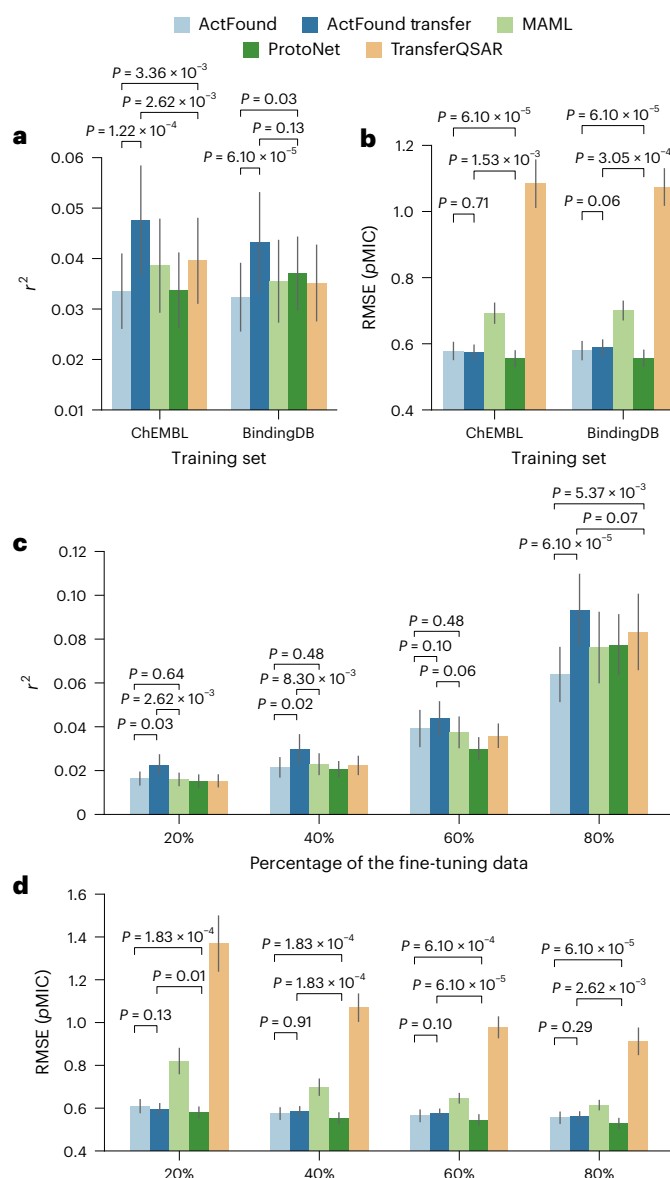

**Fig. 4 | Performance of conventional models on the NPs dataset. a,b,** Bar plots comparing each model's performance on the NPs dataset in terms of (**a**) $r^2$ and (**b**) RMSE. Plots show the mean ± s.e.m. performance values across $n = 14$ assays used to fine-tune the ChEMBL pretrained models and the BindingDB pretrained models. Before plotting, the performance values for the assays were averaged across each shot setting used to fine-tune the models. **c,d,** Bar plots comparing the models' performances when 20%, 40%, 60% and 80% of the assays were used for fine-tuning in terms of (**c**) $r^2$ and (**d**) RMSE. Plots show the mean ± s.e.m. performance values across the $n = 14$ assays used to fine-tune the pretrained models. Statistical significance between the performance of ActFound, ActFound Transfer and the best comparison approach was assessed using a one-sided Wilcoxon test.

compounds will be similar to one another. However, one disadvantage to this approach, and what we believe is causing such a range in performance for the NPs dataset, is that if the assay does not contain similar compounds, the pairwise learning function will cause large errors. In the original publication, Feng et al.[10] removed what they called 'orphan compounds' from the assays. These orphan compounds are those that have a Tanimoto similarity less than 0.2 to the other compounds in the assay. When we performed the same procedure, none of the assays had enough compounds left for fine-tuning. This is likely due to how

we defined our 'assays'. Although Porras et al. provided references in the NPs dataset, there were not enough bioactivities per bacterium in each reference to fine-tune the model. Therefore, we combined compounds from multiple references to treat each bacterial strain as its own assay, which likely led to many dissimilar compounds in each assay. Because removing the orphan compounds left only one assay available for fine-tuning, we decided not to remove them. Additionally, further analysis showed the NPs dataset having a large number of scaffolds that were not present in the training sets, as well as having more scaffold diversity than the training sets (Supplementary Tables 2 and 3). We acknowledge that this likely played a part in how well Act-Found was able to perform on the NPs dataset. However, it also reveals an important limitation of the ActFound method. NP datasets will often lack closely related pairs of compounds, as NPs are highly diverse[26], and congeners of a primary product may be difficult to discover and isolate without the use of specialized techniques[27]. This limitation may also pose a problem for high-throughput screening assay datasets that use compound libraries assembled for compound diversity over preliminary SAR, as is sometimes the case[28].

The original paper found a correlation between a small loss value for the first optimization and a large $r^2$ value. This was identified as a way to determine how well ActFound will perform on the fine-tuning assay because assays with small loss values will likely result in high $r^2$ values. However, we did not find this correlation to hold on the NPs dataset. Although most of the assays had larger loss values, even the assays with small loss values had small $r^2$ values (Fig. 3b).

## Evaluation against other state-of-the-art models

In addition to ActFound and ActFound Transfer, we also used the NPs dataset to fine-tune three other conventional models: MAML, ProtoNet and TransferQSAR. MAML and ProtoNet are meta-learning models, whereas TransferQSAR is a transfer learning variant of MAML. However, none of the models incorporate pairwise learning to learn the relative bioactivities as ActFound does. ActFound Transfer outperformed all other models with both the ChEMBL and BindingDB versions, but Act-Found performed the worst (Fig. 4a,b), a trend that holds when identical references or similar assays are moved (Supplementary Figs. 7 and 8). Even though we found ActFound to perform worse on the NPs dataset compared to the results in the original publication, ActFound and Act-Found Transfer have a higher $r^2$ value than the other three models when the fewest compounds were used for fine-tuning (Fig. 4c,d and Supplemental Fig. 9). This is indicative of ActFound's ability to quickly adapt to new assays with only a small number of fine-tuning compounds. Additionally, at this setting, the median number of compounds used for fine-tuning was nine, which is a smaller shot setting than was studied in the original paper. At each shot setting, ActFound Transfer outperformed all other models. The performance of ActFound was more varied, with it having a higher $r^2$ value than the non-ActFound variants at the 20% and 60% shot settings but having a lower $r^2$ value than most of the models at the 40% shot setting and all of the models at the 80% shot settings. Although we identified disadvantages to using pairwise learning with our dataset, these results also indicate the effectiveness of utilizing pairwise learning to learn the relative bioactivity values.

## Discussion and conclusion

To investigate ActFound's reusability, we fine-tuned the model on an antibacterial NPs dataset, studying its performance on a range of shot settings. With the availability of Feng et al.'s[10] Google Colab, we found ActFound to be easy to use and fine-tune on our dataset. In contrast to the results of the original publication, ActFound Transfer was found to perform better than the meta-learning variant of ActFound. We also found both variants of ActFound to have diminished performance compared to the cross-domain setting in the original paper. This was likely due to the incompatibility of ActFound with the NPs dataset, as the assays in this dataset contained dissimilar compounds. This meant

the pairwise learning function of ActFound was not able to be fully advantageous for our dataset. Another possible explanation for the relatively low accuracy is that NPs have different chemical properties than synthetic compounds, which likely make up most of the training data. However, given that the *t*-SNE analysis and Tanimoto similarity distribution show that the chemical space of the NP dataset overlaps with the training set, we believe that the lack of suitable pairs of compounds for pairwise learning contributes more to lowering the accuracy. Despite the poorer performance on our chosen dataset, we found both variants of ActFound to perform better than other state-of-the-art models at the lower shot settings. Therefore, we believe ActFound to be a very useful framework for those who do not have enough labelled data to train a task-specific DL model—especially those whose datasets consist of structure–activity relationship studies, as these datasets will contain the bioactivities of similar compounds, which will increase the capabilities of the pairwise learning function. However, it is important to note that the more challenging problem of accurate activity prediction for compounds in unassayed areas of chemical space remains unsolved by ActFound and other DL methods.

## Methods

### Dataset preparation

The NPs dataset used in this Rreusability Report was obtained from ref. 15. To prepare the dataset for fine-tuning, we followed a pipeline similar to that used by Feng et al.[10] when evaluating ActFound on two kinase inhibitor datasets, KIBA[29] and Davis[30]. In this cross-domain setting, they considered each kinase to be its own assay. Similarly, we considered each bacterial strain as a separate assay. The NPs dataset contains 1,439 growth-inhibitory values of 472 unique compounds against 115 bacterial strains. We considered resistant strains and subspecies as separate assays from the original strain. Assays with fewer than 20 compounds were removed. If an assay contained duplicate compounds, the growth-inhibitory values were averaged across the duplicated compounds. Additionally, we considered only compounds with minimum inhibitory concentration (MIC) values and ug ml$^{-1}$ units. This left us with 14 assays with an average of 64 compounds per assay.

### Model fine-tuning

All of the foundation models (ActFound, ActFound Transfer, MAML, ProtoNet and TransferQSAR) trained by Feng et al. were obtained from their figshare at https://figshare.com/articles/dataset/ActFound_data/24452680 (ref. 31). We fine-tuned the models using the public code of ActFound from its GitHub repository at https://github.com/BFeng14/ActFound.git (ref. 32). During the fine-tuning process, the hyperparameters and architecture for each model were the same as used by Feng et al.[10]. The input for the models were 2,048-dimensional Morgan fingerprints, which were computed using RDKit[33] and the negative log of the MIC values $p(\text{MIC}) = -\log_{10}(\text{MIC})$.

Each model was fine-tuned using the 8-, 16-, 32-, 64- and 128-shot settings. Additionally, each model was fine-tuned on different proportions of the data. During this fine-tuning stage, we used 20%, 40%, 60% and 80% of the assay data for fine-tuning. Each assay was randomly split 40 times into the fine-tuning and testing sets, and the models were fine-tuned with each random split. The results of the models are an average across each iteration.

### *t*-SNE

To study the similarities between the compounds in the ChEMBL training set and the NPs dataset, we performed a *t*-SNE analysis using scikit-learn's *t*-SNE[34]. A *t*-SNE reduces the dimensionality of the Morgan fingerprints from 2,048 to 2. We used the default values for each parameter except the distance metric, which we set to 'jaccard'. Because the Jaccard distance, or the Tanimoto distance, is equal to 1 − Tanimoto similarity, the distance between points is directly related to the similarity between compounds. The ChEMBL training set is large (1.4 million

datapoints), and running a *t*-SNE on the entire dataset would be computationally expensive. Therefore, to speed up the computation, we opted to randomly select 50% of the training set to perform the *t*-SNE.

## Data availability

The original antibacterial NPs dataset used in this report is available in ref. 15. Our processed dataset is available via figshare at https://doi.org/10.6084/m9.figshare.30334318.v2 (ref. 35).

## Code availability

The original ActFound code is available via GitHub at https://github.com/BFeng14/ActFound.git (ref. 32). All of the foundation model checkpoints are available via figshare at https://figshare.com/articles/dataset/ActFound_data/24452680 (ref. 31). Our modified code, along with the split speeds we used to fine-tune the NPs dataset on the models, is available via GitHub at https://github.com/caitlinbutt04/Actfound_reusability_report.git (ref. 36).

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

## Acknowledgements

Research reported in this publication was supported by the National Institute of General Medical Sciences under grant no. R35GM146987 (A.S.W.). The content is solely the responsibility of the authors and does not necessarily represent the official views of the National Institutes of Health. We would like to acknowledge Vanderbilt's ACCRE computing cluster for computational resources.

## Author contributions

C.M.B. performed all evaluations of the model, wrote and modified the code for this work, analysed the data and wrote the paper and Supplementary Information. A.S.W. conceived and supervised the study and edited the paper and Supplementary Information.

## Competing interests

The authors declare no competing interests.

## Additional information

**Correspondence and requests for materials** should be addressed to Allison S. Walker.

