## [Peer Review File · Nature Machine Intelligence]

Reusability Report: evaluating the performance of a meta-learning foundation model on predicting the antibacterial activity of natural products

Corresponding Author: Dr Allison Walker

Version 0:

Reviewer comments:

Reviewer #1

(Remarks to the Author)

Nature Machine Intelligence - NATMACHINE-TEL-A25062779

The paper reuses ActFound, a pairwise meta-learning foundation model for bioactivity prediction, by fine-tuning it on a literature-curated NP antibacterial dataset (14 assays; MIC→p(MIC) targets; Morgan fingerprints). It compares ActFound and ActFound-Transfer to MAML, ProtoNet, and a transfer-learning baseline across few-shot and percentage-of-assay settings. The study tackles an important question—whether a bioactivity foundation model generalises to antibacterial NPs—and implements a reasonable fine-tuning protocol. However, scientific soundness and reproducibility are weakened by (i) unaddressed data-leakage risks, (ii) insufficient distributional analysis of NP vs. pre-training data, (iii) several unsubstantiated claims about performance trends, and (iv) incomplete sharing of fine-tuned artefacts. These issues undercut the strength of the conclusions and should be addressed. Below are the details regarding these concerns:

1) NP's tend to have a different molecular size and atomic content distribution compared to most synthetic compounds. Therefore, this should be one of the considerations while doing this experiment. Please compare data distribution from different perspectives and discuss how this might affect the performance of the model, which was pre-trained on one distribution and then fine-tuned and tested on another. The existing t-SNE analysis is far from being sufficient in this manner. The data should be analysed and compared from multiple perspectives (e.g., scaffold/class distributions; size (MW), cLogP, TPSA, HBD/HBA; Bemis–Murcko scaffold entropy; Tanimoto similarity histograms to nearest neighbors in ChEMBL/BindingDB; Fréchet ChemNet Distance or MMD on fingerprints; assay-level label-range distributions) to make a more informed judgment.

2) Authors explicitly did not de-duplicate molecules/assays across NP data vs. ChEMBL; this can inflate performance. At minimum, match by InChIKey/SMILES (standardized), remove overlaps, and re-run; check assay-level overlaps by title/target/organism/endpoint.

Authors stated: "we did not attempt to identify any identical assays between the datasets or exclude molecules potentially seen during training. We acknowledge this could have caused data leakage that inflated the performance of the model pre-trained on ChEMBL assays. Given that this dataset was the result of manual literature curation, it is unlikely that the exact data in this dataset was deposited into ChEMBL."

Ok, the explanation is plausible, but still checking this, making sure is essential. If there are mutual molecules in between, the molecules potentially seen during training should be removed.

3) Authors attribute counter-intuitive trends (16-shot best; degradation at 64/128 shots) to only four large assays dominating high-shot settings, yet do not provide per-assay shot curves. The authors should provide shot-based plots for these four assays separately, then we can see whether an increasing number of finetuning samples increases the performance as desired.

4) authors stated: "ActFound and ActFound Transfer tended to have a larger r^2 value than the other three models when fewer compounds were used for fine-tuning (Fig 4c,d)."

This is not evident from the figure; maybe it is ok for ActFound Transfer, but definitely not for ActFound. Also, this:

"When the number of fine-tuning compounds is increased, both variants of ActFound perform on par with the other models"

Again, not exactly true for ActFound. Its performance relative to the compared methods does not change much when we take r^2 into account. Please revise this part to present clear, precise statements.

5) From Discussion and Conclusion: "Especially for those whose datasets consist of structure-activity relationship studies as these datasets will contain the bioactivities of similar compounds which will increase the capabilities of the pairwise learning function."

Yes, but this is not a critical problem in the literature. We already have well-performing models before ActFound for the task of predicting the activities of compounds that are significantly similar to those in the training dataset. The real problem is when you have structurally different compounds to test.

To sum up, the discussion suggests ActFound is especially useful for SAR-rich datasets with similar compounds; however, this problem is comparatively easier and already addressed by prior methods. The harder, structurally diverse generalization should be emphasized, with scaffold-split evaluations and OOD tests.

6) Please conduct a statistical analysis to evaluate the performance differences between methods.

7) In the main text, code and pretrained checkpoints are linked to the original ActFound resources, but there is nothing about fine-tuned checkpoints, split seeds, and exact preprocessing scripts. I've observed that authors shared their GitHub repo only in the manuscript system. Why is this information not shared in the Code Availability section of the article? It is hard to understand the reasoning here. Please share this information in the manuscript.

(Remarks on code availability)

Reviewer #2

(Remarks to the Author)

In this reusability report, the authors apply the ActFound compound bioactivity foundation model recently published in Nature Machine Intelligence by Feng et al. to another problem, that of predicting antibacterial activity of natural products (NPs). Performance of ActFound is relatively poor (low r^2 values), but better than other methods tried. The poor performance is reported to depend on the difference in composition between the original training data and the specific small NP data sets, with far less similar samples. Overall, the paper provides readers with relevant insights into the (limitations on) use of ActFound for such cases.

Some limitations of the approach and the current report are:

1. Potential overlap between training and test sets. This is acknowledged (77-81), but it is not clear why this could not be prevented or at least investigated. The authors write it is "unlikely that the exact data in this dataset was deposited into ChEMBL": can this not be verified? I doubt whether performance relative to other methods is influenced by this, but it could influence the absolute performance measures.
2. Performance of ActFound is compared to that of other methods, but only those provided by Feng et al. It would be good for readers to know the state-of-the-art of NP antibacterial activity prediction, to evaluate the potential of using ActFound – ideally by using other predictors on this data, alternatively by providing an overview of performances as reported in literature.
3. Performance measure plots lack errors bars (as in the Feng et al. paper), which makes it hard to interpret performance differences.

Minor suggestions:

20 Best not start a sentence with "Especially".

48 Introduce abbreviation MAML.

60 I think dereplication is a means of avoid rediscovery; rephrase?

62 "antibacterial prediction" -> "antibacterial activity prediction"

66 Sentence a bit cumbersome (2x "fine-tuning"), perhaps rephrase?

83 (and further): "bacteria strain" -> "bacterial strain"

99 "were the four assays with" -> "yielded" (or "gave")

114 "bacteria" -> "bacterium"

(Remarks on code availability)

The code is available on Github with a brief descriptive README, linked to 5 Colab notebooks for reproducing the analyses reported in the paper. I could get some of these to work, some not (e.g. the first one stopped with an error). The Github page also provides links to Colab notebooks that should help users to fine-tune ActFound to their own dataset (not discussed in the paper, as far as I could see). I have not deeply inspected the code, but the notebooks seem properly programmed.

Version 1:

Reviewer comments:

Reviewer #1

(Remarks to the Author)

I thank authors for properly addressing all of the issues I raised during the first round of review.

(Remarks on code availability)

Reviewer #2

(Remarks to the Author)

I am satisfied with the authors' responses and changes to the manuscript, which I think have helped improve it.

(Remarks on code availability)

I previously checked the code, and spot checked the few remaining errors. The code looks good and is easily accessible in Colab notebooks.

Reusability Report: evaluating the performance of a meta-learning foundation model on predicting the antibacterial activity of natural products

Responses to Reviewers in red:

Reviewer #1:

1) NP's tend to have a different molecular size and atomic content distribution compared to most synthetic compounds. Therefore, this should be one of the considerations while doing this experiment. Please compare data distribution from different perspectives and discuss how this might affect the performance of the model, which was pre-trained on one distribution and then fine-tuned and tested on another. The existing t-SNE analysis is far from being sufficient in this manner. The data should be analysed and compared from multiple perspectives (e.g., scaffold/class distributions; size (MW), cLogP, TPSA, HBD/HBA; Bemis–Murcko scaffold entropy; Tanimoto similarity histograms to nearest neighbors in ChEMBL/BindingDB; Fréchet ChemNet Distance or MMD on fingerprints; assay-level label-range distributions) to make a more informed judgment.

We added additional analysis of the molecular properties to assess the overlap between natural compounds and ChEMBL/BindingDB, which likely also contain some natural products. Box plots comparing the distributions of MW, logP, TPSA, HBD/HBA, and label distributions can be found in Supplementary Fig 2, with summary statistics reported in Supplementary Table 1.

The Tanimoto similarity histograms can be found in Supplementary Figure 1. In the paper we updated our t-sne statement to include the following: “To determine the extent of overlap, we computed the Tanimoto similarities between compounds in the two datasets and found 248 identical compounds (Supplementary Fig 1).”

The FCD, MMD, and scaffold novelty between the NPs dataset and the training sets can be found in Supplementary Table 2. The SSE of the top 5, top 10, and top 50 most populated scaffolds in each dataset can be found in Supplementary Table 3. Updated statement in paper: “Additionally, further analysis showed the NPs dataset having a large number of novel scaffolds that were not present in the training sets, as well as having more scaffold diversity compared to the training sets (Supplementary Table 2,3). We acknowledge that this likely played a part in how well ActFound was able to perform on the NPs dataset. However, it also reveals an important limitation of the ActFound method. NP datasets will often lack closely related pairs of compounds, as natural products are highly diverse and congeners of a primary product may be difficult to discover and isolate without the use of specialized techniques.”

2) Authors explicitly did not de-duplicate molecules/assays across NP data vs. ChEMBL; this can inflate performance. At minimum, match by InChIKey/SMILES (standardized), remove overlaps, and re-run; check assay-level overlaps by title/target/organism/endpoint.

Authors stated: "we did not attempt to identify any identical assays between the datasets or exclude molecules potentially seen during training. We acknowledge this could have caused data leakage that inflated the performance of the model pre-trained on ChEMBL assays. Given that this dataset was the result of manual literature curation, it is unlikely that the exact data in this dataset was deposited into ChEMBL."

Ok, the explanation is plausible, but still checking this, making sure is essential. If there are mutual molecules in between, the molecules potentially seen during training should be removed.

We identified similar compounds between the fine-tuning and training sets by removing compounds in the NPs dataset that had a Tanimoto similarity of 1 to a compound in the ChEMBL dataset. We then fine-tuned ActFound using this dataset. We followed the same procedure for overlap between the BindingDB datasets. The results of this can be found in Supplementary Figure 1. Considering the ChEMBL dataset included 1.6 million different assays, it is possible the identified compounds had bioactivities other than growth inhibitory activities present in the training datasets.

Since we combined compounds from multiple references to create our assays, we could not identify overlap based on ChEMBL assay ids alone. We used the followed procedure stated in our Supplementary file to address assay overlap:

"The antibacterial NPs dataset is a result of a manual literature review and contains the references from the original papers that the NPs were curated from. Using the ChEMBL webresource client⁷, we used the DOIs of the references to determine if any of the documents had been deposited into ChEMBL. We found nine references that had information deposited into ChEMBL. We removed the compounds associated with these nine references (35 total compounds) and fine-tuned ActFound, ActFound Transfer, MAML, ProtoNet, and TransferQSAR. Each model was fine-tuned using 20%, 40%, 60%, and 80% of the assay data for fine-tuning. Each assay was randomly split 40 times into the fine-tuning and testing sets, and the models were fine-tuned with each random split. The results of the models are an average across each iteration.

To search for identical growth inhibitory assays, we used the ChEMBL webresource client to search ChEMBL for assays of the same molecule/organism pairs that are in the antibacterial NPs dataset. We considered the assay a match and removed it from the NPs dataset if it had the same molecule/organism pair, the `bio_label` was equal to 'organism-based format', the standard type was equal to 'MIC', the standard units equal to 'ug/mL' and the standard relation equal to '='. Using these criteria, we identified and removed 324 molecule/organism pairs in the NPs dataset. We fine-tuned each model using the same procedure as when we fine-tuned after removing identical references (Supplementary Figure 3,7-8)."

Updated statement in manuscript: "Given that this dataset was the result of manual literature curation, it is unlikely that the exact assays in this dataset were deposited into ChEMBL. We investigated the overlap between the bioactivities in the NPs dataset and the ChEMBL database and identified 324 instances where identical molecules were tested in similar assays across the

two datasets. When fine-tuning, we found that removing these bioactivities from the dataset had no significant impact on the performance of ActFound (Supplementary Fig. 3). Due to this, we chose to include the overlapping bioactivities in the fine-tuning dataset.”

3) Authors attribute counter-intuitive trends (16-shot best; degradation at 64/128 shots) to only four large assays dominating high-shot settings, yet do not provide per-assay shot curves. The authors should provide shot-based plots for these four assays separately, then we can see whether an increasing number of finetuning samples increases the performance as desired.

We created the per-assay shot figures for the top four largest assays, which can be found in Supplementary Figure 5. Updated statement in manuscript: “This caused the average performance of ActFound across the assays to decrease as the shot setting increased. The performance of ActFound on the four largest assays across the 8- to 128-shot settings showed the r^2 value did increase as the shot-setting increased, following the expected trend (Supplementary Fig 5).”

4) authors stated: "ActFound and ActFound Transfer tended to have a larger r^2 value than the other three models when fewer compounds were used for fine-tuning (Fig 4c,d). "

This is not evident from the figure; maybe it is ok for ActFound Transfer, but definitely not for ActFound. Also, this:

"When the number of fine-tuning compounds is increased, both variants of ActFound perform on par with the other models"

Again, not exactly true for ActFound. Its performance relative to the compared methods does not change much when we take r^2 into account. Please revise this part to present clear, precise statements.

We updated our statements in the manuscript to reflect this. We should note that the figures have been updated since we re-fine-tuned the models to save the split seeds. We now see ActFound performing the worst at the higher shot settings, so the ‘on-par’ statement was removed

Updated statement in manuscript: “ActFound Transfer outperformed all other models with both the ChEMBL and BindingDB versions, but ActFound performed the worst (Fig 4a,b), a trend that holds when identical references or similar assays are moved (Supplementary Fig 7, 8). Even though we found ActFound to perform worse on the NPs dataset compared to the results in the original publication, ActFound and ActFound Transfer have a higher r^2 value than the other three models when the fewest number of compounds were used for fine-tuning (Fig 4c,d, Supplemental Fig 9). This is indicative of ActFound’s ability to quickly adapt to new assays with only a small number of fine-tuning compounds. Additionally, at this setting, the median number of compounds used for fine-tuning was 9, which is a smaller shot setting than was studied in the original paper. At each shot setting, ActFound Transfer outperformed all other models. The

performance of ActFound was more varied, with it having a higher r^2 value than the non-ActFound variants at the 20% and 60% shot settings, but having a lower r^2 value than most of the models at the 40% shot setting and all of the models at the 80% shot settings.”

5) From Discussion and Conclusion: "Especially for those whose datasets consist of structure-activity relationship studies as these datasets will contain the bioactivities of similar compounds which will increase the capabilities of the pairwise learning function."

Yes, but this is not a critical problem in the literature. We already have well-performing models before ActFound for the task of predicting the activities of compounds that are significantly similar to those in the training dataset. The real problem is when you have structurally different compounds to test.

To sum up, the discussion suggests ActFound is especially useful for SAR-rich datasets with similar compounds; however, this problem is comparatively easier and already addressed by prior methods. The harder, structurally diverse generalization should be emphasized, with scaffold-split evaluations and OOD tests.

We fine-tuned all of the models on a scaffold split following the procedure stated in our supplementary file: “We followed the ‘realistic’ split procedure established by Martin et al. which uses hierarchical clustering, a Tanimoto similarity-based clustering method⁸. We modified their implementation so that the clusters were randomized and then assigned to the fine-tuning and testing sets. Each assay was split 40 times into the fine-tuning and testing sets, and each model was fine-tuned with each scaffold split. We used 20%, 40%, 60%, and 80% of the assay data for fine-tuning. The results of the models are an average across each iteration (Supplementary Figure 6,9).”

We did not see significant differences between the scaffold and random splits, likely because the NP dataset is diverse enough that there is not a significant reduction in fine tuning and test compound similarity when using a scaffold split compared to a random split.

We added the following statement in manuscript to describe the scaffold split results: “We also examined fine-tuning the model on a scaffold split, which was defined as a ‘realistic split’ by Feng et al. In a scaffold split, the molecules within each assay were split so that the compounds within the fine-tuning set were dissimilar to those in the testing set. We found there to be no significant difference in r^2 values for ActFound or ActFound Transfer when a scaffold split was performed instead of a random split (Supplementary Fig 6).”

We also added the following text to the discussion and conclusion to stress that predictions for structurally distinct compounds remains difficult problem: “However, it is important to note that the more challenging problem of accurate activity prediction for compounds in unassayed areas of chemical space remains unsolved by ActFound and other deep learning methods.”

6) Please conduct a statistical analysis to evaluate the performance differences between methods.

We added error bars and conducted a one-sided Wilcoxon test between ActFound and ActFound Transfer, ActFound and the best non-ActFound variant, and ActFound Transfer and the best non-ActFound variant. The updates can be seen in the figures.

7) In the main text, code and pretrained checkpoints are linked to the original ActFound resources, but there is nothing about fine-tuned checkpoints, split seeds, and exact preprocessing scripts. I've observed that authors shared their GitHub repo only in the manuscript system. Why is this information not shared in the Code Availability section of the article? It is hard to understand the reasoning here. Please share this information in the manuscript.

We updated the Code Availability section to add our GitHub repo: "Our modified code, along with the split speeds we used to fine-tune the NPs dataset on the models, is available via GitHub at https://github.com/caitlinbutt04/Actfound_reusability_report.git."

Reviewer #2:

1. Potential overlap between training and test sets. This is acknowledged (77-81), but it is not clear why this could not be prevented or at least investigated. The authors write it is "unlikely that the exact data in this dataset was deposited into ChEMBL": can this not be verified? I doubt whether performance relative to other methods is influenced by this, but it could influence the absolute performance measures.

This was identified by Reviewer #1 as well. See our response to Reviewer #1, Comment #2 to see how we addressed this concern.

2. Performance of ActFound is compared to that of other methods, but only those provided by Feng et al. It would be good for readers to know the state-of-the-art of NP antibacterial activity prediction, to evaluate the potential of using ActFound – ideally by using other predictors on this data, alternatively by providing an overview of performances as reported in literature.

To our knowledge, there is not an established state-of-the-art foundation model for natural products antibacterial activity prediction, or natural product activity prediction methods in general. Many current models trained for antibacterial prediction contain NPs in their training datasets and therefore may be applicable to both synthetic and natural products, which was one of the questions we sought to address with this reusability approach. The only NP foundation model we are aware of is NaFM (Ding et al. arXiv. 2025), but this model is pretrained only on NP structure not activity so we do not believe it is a good comparison for ActFound, as it will likely perform worse without significant data for fine tuning. The reason we did not test it is that it is also hard coded to only work with specific datasets, none of which are antibacterial activity,

and would therefore require alteration to work with our dataset. If there is a foundation model for NP antibacterial prediction that we have overlooked, we would be happy to add a comparison to that model.

3. Performance measure plots lack errors bars (as in the Feng et al. paper), which makes it hard to interpret performance differences.

We added error bars and conducted a one-sided Wilcoxon test between ActFound and ActFound Transfer, ActFound and the best non-ActFound variant, and ActFound Transfer and the best non-ActFound variant. The updates can be seen in the figures.

Minor suggestions:

20 Best not start a sentence with "Especially".

We have updated the sentence to:

"Our results establish ActFound as a useful foundation model for the bioactivity prediction of tasks with limited data, particularly on datasets that contain the bioactivities of similar compounds."

48 Introduce abbreviation MAML.

We have added the following text:

"*k*-nearest neighbors- model-agnostic meta-learning (kNN-MAML)"

60 I think dereplication is a means of avoid rediscovery; rephrase?

We have clarified this sentence, it now reads:

"However, antibacterial NPs have historically been plagued by the need for a dereplication process, to avoid frequent rediscovery of known NPs¹⁷."

62 "antibacterial prediction" -> "antibacterial activity prediction"

We added 'activity' to this phrase

66 Sentence a bit cumbersome (2x "fine-tuning"), perhaps rephrase?

Rephrased to: "We investigated the use of the few-shot setting, fine-tuning the models on differing amounts of NPs within the dataset. The shot-settings ranged between 8 and 128 fine-tuning compounds."

83 (and further): "bacteria strain" -> "bacterial strain"

Updated “bacteria strain” to “bacterial strain”

99 “were the four assays with” -> “yielded” (or “gave”)

Changed “were the fours assays with” to “yielded”

114 “bacteria” -> “bacterium”

Changed “not enough bioactivities per bacteria” to “not enough bioactivities per bacterium”

Reviewer #2 (Remarks on code availability):

The code is available on Github with a brief descriptive README, linked to 5 Colab notebooks for reproducing the analyses reported in the paper. I could get some of these to work, some not (e.g. the first one stopped with an error). The Github page also provides links to Colab notebooks that should help users to fine-tune ActFound to their own dataset (not discussed in the paper, as far as I could see). I have not deeply inspected the code, but the notebooks seem properly programmed.

We fixed the error in the Colab notebook, so they are all now working. We updated the Code Availability section to add our GitHub repo: “Our modified code, along with the split speeds we used to fine-tune the NPs dataset on the models, is available via GitHub at https://github.com/caitlinbutt04/Actfound_reusability_report.git.”

As seen in their comments pasted below, neither reviewer had any additional concerns. We thank the reviewers for their time in reviewing and helping us improve the initial submission and in reading our revision.

Reviewer #1 (Remarks to the Author):

I thank authors for properly addressing all of the issues I raised during the first round of review.

Reviewer #2 (Remarks to the Author):

I am satisfied with the authors' responses and changes to the manuscript, which I think have helped improve it.

Reviewer #2 (Remarks on code availability):

I previously checked the code, and spot checked the few remaining errors. The code looks good and is easily accessible in Colab notebooks.